# Environmental Efficacy, Climate Change Beliefs, Ideology, and Public Water Policy Preferences

**DOI:** 10.3390/ijerph18137000

**Published:** 2021-06-30

**Authors:** Erika Allen Wolters, Brent S. Steel

**Affiliations:** School of Public Policy, Oregon State University, 300 Bexell Hall, Corvallis, OR 97331, USA; bsteel@oregonstate.edu

**Keywords:** water policy, environmental efficacy, climate change

## Abstract

Water is an unpredictable and often overallocated resource in the American West, one that strains policy makers to come up with viable, and politically acceptable policies to mitigate water management concerns. While large federal reclamation projects once dominated western water management and provided ample water for large scale agricultural development as well as the urbanization of the West, water engineering alone is no longer sufficient or, in some cases, a politically acceptable policy option. As demand for water in the West increases with an ever-growing population, climate change is presenting a more challenging and potentially untenable, reality of even longer periods of drought and insufficient water quantity. The complexity of managing water resources under climate change conditions will require multifaceted and publicly acceptable strategies. This paper therefore examines water policy preferences of residents in four western states: Washington, Oregon, California, and Idaho. Using a public survey conducted in these states in 2019, we examine preferences pertaining to infrastructural, education, incentives and regulation specifically examining levels of support for varying policies based on climate change and environmental efficacy beliefs as well as geography, demographic variables, and political ideology. Results show support for all water policies surveyed, with the exception of charging higher rates for water during the hottest part of summer. The most preferred water policies pertained to tax incentives. Some variation of support exists based on gender, education, environmental values, efficacy, state residency and belief in anthropogenic climate change.

## 1. Introduction

Water in the American west has long been a contested resource. The unpredictable and often overallocated resource strains policy makers to come up with viable, and politically acceptable, policies to mitigate water management concerns. While Western water engineering to store and move water from place to place is part of the historical development of the West, the increasing challenges posed by population growth and climate change undermine the engineering solutions of the past. The federal reclamation projects developed by the U.S. Bureau of Reclamation and the U.S. Army Corp of Engineers to quell the thirst of the arid West and to provide water for agriculture and household use was very successful in facilitating western land settlement (notably the land was already occupied by Indigenous people). However, somewhat perversely, those same projects that helped to deliver water to users, produce energy, and provide the illusion of consistent and predictable water availability are now being significantly challenged to meet the needs of continued energy expansion, agricultural irrigation, household use, and industrial use while complying with environmental laws that necessitate adequate and clean water for ecosystems, species, and consumption. It has therefore driven the development of laws like state laws in California that require developers to demonstrate adequate water supply to provide for new housing developments [1]. It has also increased the use of the Endangered Species Act (ESA) and the Clean Water Act (CWA) to protect habitat and water resources for endangered or threatened species and for human consumption and use.

Water availability and distribution is intricately tied to a hope that drought conditions will end, and that more precipitation falls in the form of snow to provide both storage and subsequently water to lower elevations in the spring thus recharging surface waters. Climate change has dramatically impacted this reality, instead offering a more challenging, and untenable, reality of longer periods of drought and continued warming. Concomitantly, western water is stressed to meet the needs of existing demand, let alone new demands. Overallocation of surface waters and the increased pumping of groundwater, often at a rate faster than recharge [2], means that any efforts to provide water for current and future demand requires conservation, storage, or incentives/disincentives for water use, an inherently political decision. Additionally, impacts from depleting groundwater include less water for streams and lakes, the need for deeper wells, impacts to water quality, and land subsistence [2], further challenging policy options to mitigate water demand.

The complexity of managing water resources under climate change conditions will require multifaceted and publicly acceptable strategies. While water management is nothing new to the American West, the increased water demand for food and energy production concurrent with increased and prolonged periods of drought due to climate change, necessitates publicly acceptable policies to manage water resources in a timely manner. Understanding public policy preferences is important because Western residents are part of a social system, and political system, where public input can have direct input on policy outcomes. This paper therefore examines public water policy preferences in four western states: Washington, Oregon, California, and Idaho. Using a household survey conducted in these states (*n* = 4695), we examine preferences pertaining to infrastructural, education, incentives, and regulatory policies. Further, we examine levels of support for varying policies based on geography, demographic variables, belief in climate change, personal efficacy, informedness, and political ideology. Examining several individual variables in addition to state of residency, this paper offers a unique perspective on water policy preferences, and provides valuable insight into where there are consistently higher levels of policy support that could be leveraged into more immediate water conservation policies.

## 2. Background

All eleven western states (WA, OR, CA, ID, NV, MT, WY, UT, CO, AZ, NM) have experienced a decline in snowpack from 1955 to 2016, most prominently in Oregon and Washington [3]. The decline is consistent with general warming trends where more precipitation falls as rain rather than snow [3] leading to an overall decline in snowpack that is less reliable as spring runoff to feed into human and ecological uses. “Millions of people in the West depend on the melting of mountain snowpack for power, irrigation, and drinking water” [3]. Within each state, there will be areas of more pronounced water concerns, for example in the northern part of Idaho there is little concern over future water availability due to ample precipitation and snowpack, but in lower elevation areas water storage and availability may lessen over time [4] Water laws in three states, Oregon, Washington, and Idaho follow appropriative rights (i.e., prior appropriation), where water rights are granted to senior users. California has a dual rights system that includes both prior appropriation and riparian rights, where ownership of land next to a water source has the right to use that water. All of these water rights confer legal rights to use water. While rights acquired through prior appropriation can be sold or transferred, riparian rights transfer with any selling of the land adjacent to the water source [5]. Further, western states are experiencing some of the highest levels of population growth in the nation. In 2019, the eleven western states experienced a 38.3% population increase [6].

Both Oregon and Washington have significantly more precipitation on the western part of the state, east of the Cascade mountain range is significantly drier. For example, in Washington average precipitation from 2000–2015 west of the Cascades was 80 inches per year, compared to only 16 inches on the eastern side [7]. Northern California experiences more precipitation in the northern part of the state and in the Sierra Nevada mountain range, while southern California is more arid. Norther Idaho is far more water abundant than southern Idaho where the lower elevations are drier and see less precipitation. However, all four states are currently under some degree of drought (see Figure 1), as climate change continues to shift the precipitation regime in these states.

Agriculture extracts the most water in the West, but it also represents one of the most significant contributors to the economy as it is a multi-billion-dollar economic industry [8]. Ongoing efforts to remove dams in the west (including dams on the Snake River in Washington, and dams on the Klamath River in Oregon and California) for ecological purposes including protecting endangered species and habitat, is often in conflict with efforts to provide water storage for distribution to water users in drier summer months, to hold water in storage for times of anticipated drought and to provide flood protection.

### State Water Profiles

*California:* California uses more water than any other state in the US [9], with water needs varying depending on the region. Snowpack in the Sierra Nevada’s (northern part of the state) is the primary way the state received water, but the southern part of the state demands 80 percent of this water due to drier, arid climate and water needs [10]. About 60% of water withdrawn is for agriculture, followed by Thermoelectric production (roughly 17%) and public supply (16%) [11] California also uses a significant amount of groundwater with one-third of total water supply coming from groundwater [5] and leading to large-scale sinking of the Central Valley [12]. Lack of water increases water quality problems resulting in Governor Newsom proposing a $140 million yearly tax ($110 million from urban water districts and $30 million from the agricultural industry) to repair infrastructural deficiencies that has compounded the problem of water quality (safe drinking water) in California [13] (the proposal was rejected by the state legislature). California will continue to deal with water stress as California’s population is expected to increase from about 39.5 million in 2017 to 44 million by 2030 [14]. However, even with increasing population, Californians have reduced water use from 46 billion in 2005, to 38 billions of water per day in 2010 [15] suggesting that water conservation policies can have a positive impact on overall water conservation for the state.

*Oregon:* While surface water is relatively abundant in many months, the summer months require the most water allocation for agriculture which accounts for approximately 85% of total water use in the state [16]. Much like other Western states, this demand outweighs existing water supplies with surface water overallocated or fully allocated in summer months, with little to no surface water available for additional allocation [17]. Further, water is critical to energy production in Oregon, where 40% of electricity in the state comes from hydroelectric generation [17]. A report by the Oregon Water Resources Department (2015) projects that by 2050 water will be further stressed due to a 40% increase in population (another 1.5 million people) a 20% increase in municipal and industrial demand, a possible 9% increase in water consumption by Oregon agricultural crops, a potential 14% increase in statewide average irrigation demand and temperature changes of up to 8.5°F [16].

*Idaho:* Idaho is the third largest water consumer in the nation in 2015 [18]. Top water withdraw includes agricultural irrigation, aquaculture and public supply, with 86% of total freshwater withdraw allocated to irrigating crops [18]. Aquaculture in Idaho uses more water than in any other state [9], accounting for 11% of total water withdraws [18]. Like Oregon, hydropower is important to Idaho, providing 53% of the state’s energy portfolio [19]. Notably, the state has already seen the impacts of reduced water availability on hydroelectricity generation. While it once supplied approximately two-thirds of the state’s energy production, drought and water stress has reduced this level to about three-fifths [20].

*Washington:* Approximately 80% of state water use in Washington in 2015 was for irrigation and public supply, with agricultural irrigation comprising 59% of total water use for the state [7]. In 2003, the Washington State Legislature passed the Municipal Water Law which subsequently led to water efficiency efforts like the installation of water meters (by 2017), and reduced water loss through improved technology and distribution [21]. Since 2010, water withdraw has decreased by 13% [7]. Further, over last 15 years water use has declined in both the public and domestic sector even though population continued to rise [7] due to water conservation and water-efficient tech. These water conservation strategies will become more significant as it is anticipated that the state will continue to grow in population adding another million people to the state’s current 7.6 million residents [22].

States are well-aware of water management needs and have each set about with different strategies to address those needs. California, Idaho, and Oregon all have statewide strategies (the California Water Plan, the Idaho Comprehensive State Water Plan, and Oregon’s Integrated Water Resource Strategy), while Washington addresses water management on the county level. All of the plans identify water use and need while simultaneously attempting to provide potential management options for current and future water demand, including conservation strategies to address ecological needs as well as comply with federal laws related to water quality, sovereignty, and endangered species habitat. However, all states adhere to the water allocation laws that have solidly established the right to use water through prior appropriation or riparian water rights. Working within these confines, all four states identify the need for conservation, technology and other strategies to help conserve water quantity while meeting water use needs.

## 3. Literature Review

Support for water conservation policies can vary based on a myriad of factors, especially when considering long-term impacts. For example, mandatory policies can have a boomerang effect when restrictions ease, such as in California after mandatory water restrictions were lifted water use went back up [23]. In understanding how employing varied policy tools like conservation, infrastructural development and price discrimination techniques to address water issues it is helpful to ascertain potential public support and identify areas where more people could be nudged to support water policies by examining potential explanatory variables.

Several studies have explored the whether personal efficacy, or the belief that individual actions (either by behavior or support for policies, etc.) can have a positive impact, is a strong motivator for people. In a meta-study by Witte and Allen (2000) on public health, they found the greater the sense of efficacy correlated to stronger “attitudes, intentions, and behaviors toward the recommended response” (598) [24]. Bandura (1994) suggest that more people perceive self-efficacy, “… the higher the goal challenges people set for themselves and the firmer is their commitment to them” (3) [25]. Jugert et al. (2016) found that self-efficacy is necessary to activate collective efficacy that would lead to greater individual environmental behaviors [26]. Thaker et al. (2018) found that people who reported “high levels of perceived collective efficacy” were more likely to support water conservation policies [27]. Other studies have also found the positive impact of efficacy on support for and/or engagement in environmental policies or behaviors [28,29].

Research on sociodemographic variables and support for water conservation and environmental policies determine that women are more likely to demonstrate environmental concern [30,31,32,33], and are somewhat more likely to feel household water conservation is effective at mitigating water quantity concerns [34]. People with higher levels of formal education tend to also support water conservation policies [32]. However, age has elicited mixed results with some researchers finding younger people are more likely to engage in water conservation behaviors [35,36,37], while others [33,38] found that as older residents more likely to engage in water conservation behaviors. Income has had mixed, if any, impacts on conservation behaviors and water conservation specifically. While Cordell and Tarrant (1997) and Olli et al. (2001) find no relationship between income and conservation behaviors generally [39,40], Fielding et al. (2012) found that lower income individuals conserve more water [37], while Wolters (2014) and Trumbo and O’Keefe (2001) find that higher income households reported greater water conservation behaviors [33,41]. However, other studies [39,40] find no relationship between income and water conservation [37]. Research into the influence of political ideology on water conservation has found no relationship [33], but there is some research suggesting that more Democratic-leaning cities are potentially more likely to adopt water conservation policies [42]. Finally, research on residency, particularly rural-urban residency has not found significant difference in environmental concern, suggesting there are other variables at play [33,43,44] However, residency seems relevant in drought prone areas as residents in these areas are more likely to demonstrate concern about water [45].

Climate change beliefs are notably divisive over political ideological lines, with 87 percent of liberal Democrats believing in anthropogenic climate change compared to 27 percent of conservative Republicans [46]. Therefore, when political beliefs are made salient when asking about beliefs in anthropogenic climate change, those who identify on the political right (conservative) indicate they do not belief in anthropogenic climate change [47]. Further, McCright and Dunlap (2011) found that not only are conservatives/Republicans less likely to be in alignment with scientists over climate change, but that view became more prominent over the time period they were researching (2001–2010) [48]. This ideological division can therefore impact beliefs, values and knowledge related to climate change that have been found to be indicators of support to varying degrees for either direct or tangential policies. Milfont (2012) found that being aware of climate change increased concern pertaining to climate risks, and this concern led to an increased sense of efficacy in helping to mitigate climate change [49]. In a study by Clark and Finley (2007), they found that knowledge of climate change was significantly related to intended conservation of water [50]. Brownlee et al. (2014) found that climate change beliefs were related to concern about local drought impacts but did not find that this belief translated to local conservation beliefs [51]. Research on farmer’s beliefs in climate change and support for mitigation and adaptation by Arbuckle Jr. et al. (2013) found a strong association between farmers concerned about climate change and support for mitigation and adaptation strategies [52].

## 4. Methods and Data

In 2018 a mail and online survey was conducted in Washington, Idaho, California and Oregon. The four Western states included as case studies were selected because of their similar histories, policy-making institutions, ecosystem services, and their historic exposure to water shortages and drought. All four of the states are currently experiencing water shortage issues and drought leading to state-wide debates about appropriate water policies to address the issues. Households in these states were selected using random address-based sampling (ABS) with surveys administered following a modified version of Dillman’s (2007) tailored design methods [53]. Households first received a postcard notifying them of the survey and providing them with an option to complete the survey online. Following the first wave of the survey, a second wave of mail surveys were sent to those who did not already complete either the online or mail-in survey. All mailings were accompanied by a letter indicating the scope of the project, contact information for the principal investigator (PI) and informing potential participants of expected time to complete the survey. Each wave of surveys contained a pre-paid, first-class return mail envelope. Finally, all households receiving the survey was asked: “If available, we would prefer the person, 18 years or older, who most recently celebrated a birthday to complete the survey.”

Random ABS sampling generated 4,695 valid residential addresses for households in CA, ID, OR, and WA (CA = 1170, ID = 1175, OR = 1173, and WA = 1177. Response rate varied slightly between states (See Table 1), with most respondents preferring to complete the mail-in survey (see Table 1). Overall, in terms of representativeness, respondents from all four states were slightly older, more affluent, and had attained higher levels of formal education when compared to 2010 Census data. However, these findings are in line with survey research [54] (see Table 2).

This study was approved by the Oregon State University Institutional Research Board (IRB) approved in December, 2017. All researchers on the project were trained and certified to conduct human subjects research and were under the supervision of certified faculty. Participation in the survey was completely voluntary and consent was given by respondents by either completing the paper version of the survey and returning it in a postage pre-paid envelope, or by following the link and completing the survey online. All survey responses are stored on password and virus protected computers.

Results for independent and control variables for the combined state samples are provided in Table 2. Sociodemographic variables were measured with varied techniques. Age was measured as an open-ended question “What is your current age in years?”, providing a range from 18 years old to 98 years old. The average age of respondents being 51.6 years old. Gender was ascertained using a choice of female, male and “prefer not to say.” Only three respondents replied, “prefer not to say,” with those answering male or female coded as follows: 1 = women and 0 = men. Slightly more women responded to the survey with a mean of 0.504. Education and income were measured using a multi-categorical response choice, asking respondents to provide their highest level of formal education and their household income before taxes. Most respondents indicated that they had “Some college, no degree”, and the most common income category between $50,000–$74,999 in 2017 before taxes. Political ideology was measured using a nine-point scale with 1 = “Very Liberal” to 9 = “Very Conservative.” Mean results on political ideology show a moderate with a slight liberal leaning (4.68).

Efficacy was assessed using an index comprised of four questions with a Likert response scale ranging from 1 = “Strongly Disagree” to 5 = “Strongly Agree” The four questions all assessed personal efficacy: “I feel that my own personal behavior can bring about positive environmental change”; “I would be willing to accept cuts in my standard of living, if it helped the environment”; “I would be willing to support higher taxes, if it helped to protect the environment”; and “I would be willing to sacrifice some personal comforts in order to conserve resources.” The composite efficacy scale ranged from 4 = low efficacy to 20 = high efficacy.

The climate change variable was determined based off the question “From what you’ve heard or read, do scientists generally agree that the Earth is getting warmer because of human activity, or do they not generally agree about this?” Responses were collapsed and recoded into dichotomous categories where 1 = Yes, earth is getting warmer because of human activity and 0 = else (those who answered do not know or not due to human activity).

Dependent variables include eight questions related to water policy preferences developed by Portney et al. (2017) [55]. The questions can be categorized into four domains: infrastructure, tax incentives, regulation, and education. All eight questions started with the prompt “A number of policy options have been proposed to manage water resources. Please indicate your level of opposition or support for each of the following options.” Responses were on a 5-point Likert scale ranging from 1 = “Strongly Oppose” to 5 = “Strongly Support.” There were two questions on infrastructure “Build dams and reservoirs” and “Build pipelines to bring water from other regions”; two questions on tax incentives, “Give tax incentives for installation of water-saving equipment” and “Give tax incentives for implementing efficient irrigation systems for agriculture”; three questions on regulation, “Charge higher water rates during the hottest part of the summer”, “Charge higher water rates for high volume user”, and “Require low water use landscaping”; and one questions on education “Conduct campaigns for voluntary water conservation” (see Table 3).

## 5. Results

Table 3 reports mean scores for all eight water policy statements by state. For the two infrastructure policy statements, California respondents were significantly more likely to support the building of dams, reservoirs and pipelines to bring water from other regions ($\bar{x}$ = 3.74). This is not surprising given the state’s on-going experience with draught, large agricultural sector, and increasing demand given population growth over the decades. Idaho respondents ($\bar{x}\;$= 3.63) were the second most likely to support the building of dams and reservoirs with Oregon ($\bar{x}\;$= 3.37) and Washington ($\bar{x}\;$= 3.34) respondents slightly in favor of dams with mean scores barely above “neutral.” Support for building pipelines to bring water from other regions is not supported by Oregon respondents ($\bar{x}\;$= 2.67), and Idaho and Washington respondents were closer to “neutral” in their support ($\bar{x}\;$=3.06 and $\bar{x}\;$ = 3.19, respectively). These results are not surprising, given that there has been much public concern about California attempting to access water in the Pacific Northwest (PNW) over the years, often stoked by PNW politicians running for office [56].

There is much support for the two tax incentive water policy statements in all four states, however support is significantly stronger for respondents in Idaho, Oregon, and Washington when compared to California respondents. Mean scores for Idaho, Oregon, and Washington are all at or above 4.00, while California has a mean score of 3.75 for tax incentives for installing water-saving equipment and 3.70 for implementing efficient irrigation systems for agriculture. These results maybe the result of California respondents wary of tax policies due to their high level of tax burden compared to the other states.

For the three water policy statements concerning regulation questions including charging higher rates during summer and for high volume users, there is support for charging higher rates for high volume users but less agreement on support for higher rates during the summer. Idaho ($\bar{x}$ = 2.64) and Oregon ($\bar{x}\;$= 2.83) respondents are significantly more likely than California ($\bar{x}$ = 3.09) and Washington (3.22) respondents to oppose charging higher rates during the hottest part of summer. However, there is support in all states for charging higher rates for high volume users with Washington having the highest mean of 3.68 and Idaho having the lowest mean ($\bar{x}\;$= 3.27) of the four states. For the mandate policy of requiring low water use landscaping, there is support for the policy is all four states with the highest mean scores are found in Oregon ($\bar{x}\;$= 3.72) and Washington ($\bar{x}\;$= 3.73), followed by Idaho ($\bar{x}\;$= 3.51) and California ($\bar{x}\;$= 3.44).

The final water policy statement concerns an educational approach with the voluntary approach of “conducting campaigns for voluntary water conservation.” There is support for an education policy in all four states, with the highest level of support for voluntary water conservation is found in Oregon ($\bar{x}\;$= 4.00) and Washington ($\bar{x}\;$= 4.15), compared to California ($\bar{x}\;$= 3.74) and Idaho ($\bar{x}\;$= 3.84).

### 5.1. Multivariate Analyses

The objective of the multivariate analyses is to examine factors that drive public preferences for the eight water policies. More specifically, we assess whether individual’s sense of efficacy, climate change beliefs, political ideology, and self-assessed level of informedness can explain preferences for the infrastructure, tax incentives, regulation, and educational approaches to water policies. We also control for the effects of various sociodemographic characteristics of individuals and the state they reside in. We use ordinal regression estimates to examine the impact of the independent variables on the eight water policy statements. The models for the water infrastructure and tax incentive policy preferences are presented in Table 4 and for the regulatory and educational policy preferences in Table 5. We include a dummy variable for California to control for state residence effects in each model since the data displayed in Table 3 generally showed that Californians had differing policy preferences from the other three states. We did rotate dummy variables for all four states; however, the Californian dummy variable produced the most robust results.

All four models in Table 4 had statistically significant Chi-Square results indicating that each model provided a good statistical fit. Pseudo R-Square coefficents are also provided including Cox and Snell and Nagelkerke measures. The lowest Pseudo R-Square measures are for the building pipelines model (0.081 and 0.84, respectively) and highest for the tax incentives for water saving devices (0.220 and 0.238). While these R-Squares are somewhat low, this is typically the pattern when analyzing public opinion data [57]. For the sociodemographic variables, age only had a significant result for the building dams model where older respondents were more supportive when compared to younger respondents as expected. Gender had a significant in all four models, with women less supportive of both infrastructure policies when compared to men, and women more supportive of both tax incentive policies for water-saving equipment and efficient irrigation systems for agriculture. These findings are consistent with the literature review as women’s policy preferences are more environmentally oriented than men.

Both education and income produced statistically significant results in the same three models. Respondents with lower levels of formal education were more supportive of building pipelines when compared to more highly educated respondents. For income, those respondents with higher levels of income were more supportive of building pipelines when compared to lower income respondents. For both tax incentive models, higher income and higher educated respondents were significantly more supportive of both policies compared to lower income and educated respondents.

The next variable in the model concerns the level of self-assessed informedness respondents indicated they have concerning water policy issues. Those respondents that indicated they had lower levels of informedness were significantly more supportive of building dams than those respondents reporting higher level of informedness. While one might argue that those considering themselves more informed may have more information on the negative impacts that dams can have on the environment (fisheries, water quality, etc.), the results for the two tax incentive policies are a little more perplexing. Those respondents with lower levels of self-assessed informedness were significantly more likely to support both tax incentive policies when compared to respondents indicating higher levels.

The environmental efficacy index had a statistically significant result in all four models. As was expected from the literature review, those respondents with high levels of efficacy were more likely to oppose both infrastructure policies of building dams and pipelines, and support both tax incentive policies for water-saving equipment and more efficient irrigation systems for agriculture.

For the two value orientation variables of climate change beliefs and political ideology, both variables had statistically significant effects for three models each. Respondents who believe in human caused global warming were significantly more likely than non-believers to oppose both water infrastructure policies yet support the tax-incentive policy to encourage the installation of water-saving equipment. Climate change beliefs had no significant effect on support or opposition for tax incentives for efficient irrigation systems. Concerning political ideology, self-identified conservatives were significantly more likely than liberals to support the building of dams and reservoirs. Conservatives were also more likely to oppose both tax incentive programs while liberals were more supportive. Political ideology had no significant effect on support or opposition to the building of pipelines to bring water from other regions.

The final variable included in each model is the dummy variable for California. When controlling for sociodemographic, level of informedness, environmental efficacy, climate change beliefs and political ideology, Californians were significantly more likely to support the building of dams, reservoirs, and water pipelines, and less supportive of both tax incentive policies when compared to respondents in the three other states.

Table 5 presents ordinal regression models for the remaining four water policies including charging higher water rates during the hottest period of the summer and for high volume users. The models for voluntary and mandatory water policies of conducting campaigns for voluntary water conservation and requiring low water use landscaping are also presented in Table 5. The Chi-Square statistic for each model is significant, indicating that the specified structure constitutes an acceptable model in the statistical sense. In addition, Pseudo R-Square measures are provided for each model ranging from a low of 0.245 and 0.257 for the higher water rates volume model to a high of 0.310 and 0.333 for the voluntary water conservation model. The models in Table 5 all have higher pseudo R-Square coefficients than in Table 4, indicating that a higher level of variation is being explained when compared to those models. As with the discussion of the previous models in Table 4, we will discuss the sociodemographic control variables first.

Similar to the previous models in Table 4, age only has a significant effect in one model at the 0.001 level. Younger respondents are significantly more likely to support higher rates in summer when compared to older respondents. Age had no significant impact for the higher water rate for high volume users, voluntary conservation, and low water use landscaping policies. Gender on the other hand, had a statistically significant effect for three models. Males are more supportive of higher summer energy rates and low water landscaping policies when compared to females. Females are significantly more supportive of voluntary water conservation policy when compared to males.

The next sociodemographic variables examined are formal educational attainment and household income. Both variables are statistically significant for three models each. Those respondents with higher levels of education are more supportive of higher water rates for high volume users, higher rates in the hottest part of summer, and voluntary water conservation campaigns than those with lower levels of education. Concerning income, those with higher levels of household income are more supportive of higher water rates for high volume users and higher summer rates, and less supportive of requiring low water landscaping than households with lower levels of income. This is an interesting finding in that higher income households support higher water rates for some, but not a requirement for low water landscaping, so those who use a lot of water will need to pay for it but are not forced to engage in low water landscaping.

Turning now to the self-assessed informedness and the environmental efficacy variables, we find that informedness has a significant effect in only one model—higher rates for summer. Those respondents who consider themselves less informed are more supportive of higher summer rates than those considering themselves more informed. This is not exactly what the literature might suggest as those more informed about water policy would theoretically be more informed about drought and water shortages facing the western U.S. However, environmental efficacy did have an impact on water policy preferences as the literature would suggest. Efficacy has a statistically significant effect for all four water policy models. Those respondents with higher levels of environmental efficacy are more supportive of higher water rates for high volume users, higher rates in the hottest part of summer, more supportive of voluntary water conservation, and more supportive of required low water landscaping than those with lower levels of environmental efficacy. This especially makes sense for the voluntary water conservation campaigns as high levels of environmental efficacy would lead to one engaging in voluntary efforts to bring about changes in others’ behaviors.

For the variables of political ideology and climate change beliefs, we find that climate change belief is significant in all four models, and political ideology is significant in two models as the literature would suggest. Those respondents who believe in human-caused global warming are significantly more supportive of all four water policies than those who do not believe. Those respondents who are more liberal are more likely than conservatives to support higher rates for high volume users and requiring low water landscaping. This is consistent with the previous literature review in the paper where those people who are liberal and believe in human-caused climate change are more supportive of policies that contribute to environmental sustainability.

Finally, the dummy variable for California is statistically significant in three of the four water policy models. When compared to respondents in the other three states, Californians are more supportive of higher water rates during the hottest part of the summer, less supportive of voluntary water conservation campaigns, yet less supportive of requirements for low water use landscaping. So, the take here is that high volume users should pay more for water in the summer, but not be required to use low water landscaping. The finding that Californians are less supportive of voluntary conservation efforts may be based on some cynicism that such efforts do not work, but this would take additional research to confirm. Overall, however, as discussed above California’s history and experience with water is much more complex than the other three states leading to different perspectives than the other western states examined in this study.

### 5.2. Summary of Results

Looking at sociodemographic variables, we find that men are more likely to support building dams and are more favorable of pipelines, charging higher energy rates, require low water landscaping, and tax incentives for efficient irrigation systems. Women, and those with higher levels of formal education, are more likely to support voluntary water conservation campaigns or tax incentives for water-saving equipment. Additionally, older respondents are more supportive of requiring low water use landscaping and building dams.

Those that are less informed about water policy are more likely to support building dams and providing tax incentives for implementing efficient irrigation systems for agriculture, tax incentives for water savings and charging higher water rates in summer. People with less formal education prefer implementing efficient irrigation systems for agriculture, while those with more formal education support policies that charge higher water rates for high volume users, charging more during the summer months, and voluntary water conservation campaigns. Additionally, as income increases, so does support for building pipelines, implementing efficient irrigation systems for agriculture, and charging higher water rates for high volume users. As income decreases, there is more support for requiring low water use landscaping.

Those with lower levels of environmental efficacy were more likely to support building dams and pipelines, while those with high levels of personal efficacy were more likely to support voluntary water conservation, tax incentives for water-saving equipment, low water use landscaping, tax incentives for efficient irrigation systems, and charging higher rates for high volume users.

People who believe in human caused climate change were more likely to support voluntary water conservation campaigns, tax incentives for water-saving equipment, low water use landscaping, and charging higher rates for high volume users. Those less likely to believe in human caused climate change supported infrastructural projects like building dams and pipelines.

Regarding political ideology, people who identify as more conservative are more likely to support infrastructure, specifically building dams, while people leaning more liberal are more likely to support a multitude of water policies including supporting voluntary water conservation campaigns, tax incentives for water-saving equipment, requiring low water landscaping, providing tax incentives for efficient irrigation systems, and charging higher rates for high volume users.

Turning now to California (compared to the other three states), Californians were more likely to support infrastructure policies like building dams and pipelines and were slightly more supportive of charging higher rates during the hottest part of summer. This contrasts with WA, ID, and OR that were more favorable of voluntary water conservation campaigns, tax incentives for water-saving equipment, low water landscaping, and tax incentives for efficient irrigation. Looking broadly at categorical support for infrastructure, education, incentives and regulation there are some commonalities. In terms of infrastructure, men, people with lower environmental efficacy, those who do not believe in anthropogenic climate change, and Californians are more likely to support building dams and pipelines. Voluntary policies, i.e., conduct campaigns for voluntary water conservation found support among women, those with higher personal efficacy, people who believe in climate change, liberals, non-California residents, and people with higher levels of formal education. People preferring incentives (tax incentives) increased with income, had higher levels of personal efficacy, were less informed about water policy issues, politically more liberal, and non-California residents. Finally, people who supported regulatory policies (requiring low water use landscaping and charging higher water rates for high volume users) had higher levels of personal efficacy, believe in anthropogenic climate change, and are more liberal.

## 6. Discussion

While variation exists between water policy preferences, it is notable that there was almost uniformly at least some support for the water policies across all variables (the exception being to charge higher water rates during the hottest part of summer). This suggests that people in the American West (ID, WA, OR, and CA) at least have some awareness of water issues persistent in the region. Overall, policies that had the most support were tax incentives and voluntary water conservation campaigns. Generally, with resource conservation, voluntary measures are more desirable to mandatory measures as voluntary policies require participation and engagement of the public [58]. There is also some differentiation between perceived individual policy impacts versus infrastructural or agricultural impacts. The policy with the least support across all four states surveyed was to charge higher rates during the summer. This is not surprising as people are generally more comfortable with a set price for water, or maximum additional fee they would be willing-to-pay [59] versus a more abstract increase in water rates.

Efficacy was one of the more significant predictors of support. Low efficacy individuals supported infrastructural policies, while people with higher levels of personal efficacy supported education, tax incentives, and regulation. Relatedly, education campaigns can impact self-efficacy. When people know how their actions have an impact, they are more likely to exhibit a sense of personal efficacy to engage in that behavior [60]. In terms of policy moving forward, it is not only imperative to conduct campaigns around voluntary water conservation but draw a direct link between specific actions and water conservation.

Further, belief in anthropogenic climate change was not a consistent predictive variable across policy types. This again suggests that more information, education or knowledge of climate change, does not align with behavioral responses to mitigate climate impacts, like water scarcity. Personal efficacy pertaining to conservation of water is less related to being informed as to the reason why (climate change) and more driven by a connection between policies and direct outcomes. As Kellstedt et al. (2008) found, “when it comes to personal efficacy regarding global warming, respondents who are better informed about the issue feel less (not more) responsible for it” (122) [61]. Therefore, aligning goals and outcomes is key to policy choices and personal behaviors that reinforce direct connection between water conservation efforts and support. Tabernero and Hernández’s (2011) study on recycling behavior found that “individuals with a higher judgment of their capacity to recycle engage in more recycling behaviors, feel more satisfied with their behavior, and feel greater intrinsic motivation” (668–669) [62].

It is encouraging that there are strong levels of support for water policy in all four states. Increased water drought and water scarcity will only stress current water demand. Water conservation policies enacted in both Washington and California have illustrated the success of these policy measures as a means of decreasing overall water use, even as both states had increasing state populations, water conservation policies led to a decrease in water use from years prior to the enactment of these policies. To encourage and increase public support for water conservation policies, it is clear that a direct connection between action and outcome is established. Simply put, when people understand how a policy will affect actual change, there is the potential for more support. Future water policies then should directly connect how the policy will actualize in direct water conservation.

## 7. Conclusions

While large infrastructural water projects once provided ample water for the American West to develop both a robust agricultural economy and urban centers, it is evident that future water policies will need to focus on both conservation and distribution due to increasingly scarce water resources in the West. This study examined public preferences regarding water policies in four Western states. Findings illustrated overall support for all of the water policies (with one exception), with levels of support varying depending on residency, sociodemographics, environmental values, efficacy and political ideology. Highest levels of support were for tax incentive policies as a means of conserving water. While the transition to large-scale water conservation provides a policy challenge, it is notable that the public broadly supports water policies. This general support for water policies suggests that American’s living in the Western United States are aware of changes to water availability, including both increasing periods of drought and water scarcity, and are willing to support policies to mitigate these issues.

## Figures and Tables

**Figure 1 ijerph-18-07000-f001:**
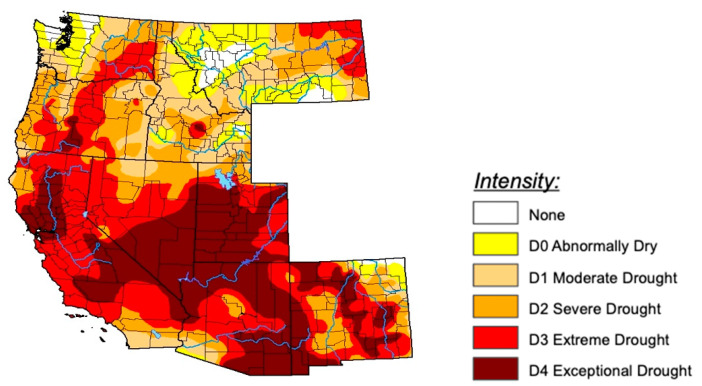
U.S. Drought Information. Source: Fusch, 2021, https://droughtmonitor.unl.edu/data/pdf/20210608/20210608_west_text.pdf (accessed on 15 January 2021). The U.S. Drought Monitor is jointly produced by the National Drought Mitigation Center at the University of Nebraska-Lincoln, the United States Department of Agriculture, and the National Oceanic and Atmospheric Administration. Map courtesy of NDMC.

**Table 1 ijerph-18-07000-t001:** Survey Response Rates.

State	Sample Size	Responses	Response Rate	% Online Return
California	1170	435	37.2%	31.7%
Idaho	1175	440	37.4%	18.9%
Oregon	1173	475	40.5%	24.2%
Washington	1177	454	38.6%	19.2%

**Table 2 ijerph-18-07000-t002:** Independent and Control Variables.

Variable Name	Variable Description	Mean
Age	Age in years (range = 18 to 98)	Mean = 51.6 s.d. = 16.83 *n* = 1796
Gender	Gender dummy variable (1 = female, 0 = male)	Mean = 0.504 *n* = 1787
Education	Formal educational attainment (1 = less than high school to 8 = postgraduate degree)	Mean = 4.80 s.d. = 1.46 *n* = 1798
Income	Household income before taxes in 2019 (1 = less than $10,000 to 10 = $200,000 or more	Mean = 5.88 s.d. = 1.80 *n* = 1772
Informed	Self-assessed informedness on water policy issues (1 = not informed to 4 = very well informed)	Mean = 2.09 s.d. = 0.94 *n* = 1799
Efficacy	Environmental efficacy index (4 = low efficacy to 20 = high efficacy)	Mean = 14.16 s.d. = 3.94 *n* = 1793
Climate Change	Climate change beliefs dummy variable (1 = earth getting warmer because of human activity, 0 = else)	Mean = 0.61 *n* = 1793
Ideology	Subjective political ideology (1 = very liberal to 9 = very conservative)	Mean = 4.68 s.d. = 1.25 *n* = 1782
California	California dummy variable (1 = California resident, 0 = else)	Mean = 24.1 *n* = 1804

**Table 3 ijerph-18-07000-t003:** Water Policy Preferences by State.

Question: A number of policy options have been proposed to manage water resources. Please indicate your level of opposition or support for each of the following options [1 = strongly oppose; 2 = oppose; 3 = neutral; 4 = support; 5 = strongly support].						
		CA	ID	OR	WA	
		Mean (s.d.) *n*	Mean (s.d.) *n*	Mean (s.d.) *n*	Mean (s.d.) *n*	F-test
Build dams and reservoirs.		3.74 (1.04) 433	3.63 (1.16) 438	3.37 (1.19) 474	3.34 (1.18) 453	12.90 *p* = 0.000
Build pipelines to bring water from other regions.		3.52 (1.12) 434	3.06 (1.27) 438	2.67 (1.18) 475	3.19 (1.29) 452	37.72 *p* = 0.000
Give tax incentives for the installation of water-saving equipment.		3.75 (1.33) 434	4.09 (0.972) 438	4.00 (1.03) 475	4.13 (0.878) 453	11.69 *p* = 0.000
Give tax incentives for implementing efficient irrigation systems for agriculture.		3.70 (1.21) 434	4.11 (0.874) 438	4.05 (0.960) 475	4.15 (0.816) 453	19.70 *p* = 0.000
Charge higher water rates during the hottest part of the summer.		3.09 (1.32) 434	2.64 (1.35) 438	2.83 (1.35) 474	3.22 (1.41) 453	16.29 *p* = 0.000
Charge higher water rates for high volume user.		3.47 (1.36) 434	3.27 (1.39) 438	3.46 (1.33) 474	3.68 (1.21) 452	7.10 *p* = 0.000
Conduct campaigns for voluntary water conservation.		3.74 (1.30) 434	3.84 (1.12) 437	4.00 (0.998) 475	4.15 (0.870) 453	12.20 *p* = 0.000
Require low water use landscaping.		3.44 (1.31) 434	3.51 (1.26) 437	3.72 (1.15) 474	3.73 (1.21) 452	6.38 *p* = 0.000

**Table 4 ijerph-18-07000-t004:** Ordinal Regression Estimates for Water Infrastructure and Tax Incentive Policy Preferences.

	Build Dams	Build Pipelines	Tax Incentives for Water-Saving	Tax Incentive Irrigation
	*Coefficient* *(S.E.)*	*Coefficient* *(S.E.)*	*Coefficient* *(S.E.)*	*Coefficient* *(S.E.)*
***Location:***				
**Age**	0.009 *** (0.003)	−0.001 (0.003)	−0.002 (0.003)	0.004 (0.003)
**Gender = 0**	0.397 *** (0.115)	0.303 *** (0.088)	−0.410 *** (0.093)	−0.385 *** (0.094)
**Education**	0.050 (0.033)	−0.069 * (0.033)	0.120 *** (0.035)	0.056 (0.035)
**Income**	−0.039 (0.026)	0.102 *** (0.026)	0.130 *** (0.027)	0.142 *** (0.028)
**Informed**	−0.155 *** (0.047)	0.051 (0.046)	−0.145 ** (0.049)	−0.257 *** (0.050)
**Efficacy**	−.043 ** (0.014)	−0.051 *** (0.014)	0.158 *** (0.015)	0.134 *** (0.015)
**Climate Change = 0**	0.397 *** (0.115)	0.435 *** (0.114)	−0.456 *** (0.119)	−0.111 (0.120)
**Ideology**	0.140 *** (0.024)	−0.036 (0.023)	−0.059 * (0.025)	−0.060 * (0.025)
**California = 0**	−0.517 *** (0.105)	−0.768 *** (0.103)	0.450 *** (0.107)	0.659 *** (0.108)
***Threshold:***				
**Variable = 1**	−2.075 *** (0.376)	−2.006 *** (0.367)	−1.130 ** (0.390)	−2.001 *** (0.404)
**Variable = 2**	−1.133 ** (0.371)	−0.879 * (.363)	−0.100 (0.383)	−0.488 (0.386)
**Variable = 3**	0.708 * (0.370)	0.550 (.363)	1.096 ** (0.383)	0.468 (0.384)
**Variable = 4**	1.947 *** (0.373)	1.668 *** (0.365)	3.164 *** (0.390)	2.832 *** (0.391)
***n*=**	1711	1711	1712	1712
**Chi-Square=**	239.167 ***	144.629 ***	428.008 ***	294.625 ***
**Cox and Snell=**	0.130	0.081	0.220	0.157
**Nagelkerke=**	0.137	0.084	0.238	0.172

* *p* ≤ 0.05; ** *p* ≤ 0.01; *** *p* ≤ 0.001.

**Table 5 ijerph-18-07000-t005:** Ordinal Regression Estimates for Water Rates and Regulatory Policy Preferences.

	Higher Rates High Volume	Higher Rates Summer	Voluntary Water Conservation	Low Water Landscaping
	*Coefficient* *(S.E.)*	*Coefficient* *(S.E.)*	*Coefficient* *(S.E.)*	*Coefficient* *(S.E.)*
***Location:***				
**Age**	−0.004 (0.003)	−0.012 *** (0.003)	0.001 (0.003)	0.005 (0.003)
**Gender = 0**	−0.094 (0.089)	0.252 ** (0.089)	−0.253 ** (0.093)	0.180 * (0.090)
**Education**	0.065 * (0.033)	0.157 *** (0.033)	0.125 *** (0.035)	0.059 (0.033)
**Income**	0.091 *** (0.026)	0.089 *** (0.026)	0.005 (0.027)	−0.132 *** (0.027)
**Informed**	0.000 (0.047)	−0.096 * (0.047)	−0.086 (0.049)	0.015 (0.048)
**Efficacy**	0.177 *** (0.014)	0.207 *** (0.015)	0.249 *** (0.016)	0.203 *** (0.015)
**Climate Change = 0**	−0.460 *** (0.114)	−0.421 *** (0.114)	−0.663 *** (0.119)	−0.591 *** (.116)
**Ideology**	−0.077 *** (0.024)	−0.030 (0.024)	−0.026 (0.025)	−0.166 *** (0.024)
**California = 0**	0.039 (0.104)	0.231 * (0.104)	−0.394 *** (0.108)	0.472 *** (0.105)
***Threshold:***				
**Variable = 1**	0.386 (0.369)	1.431*** (0.369)	−0.370 (0.389)	−1.278 *** (0.404)
**Variable = 2**	1.385 ** (0.369)	2.818 *** (0.374)	0.692 (0.384)	−0.155 (0.386)
**Variable = 3**	2.460 *** (0.373)	3.840 *** (0.379)	2.301 *** (0.387)	1.328 *** (0.373)
**Variable = 4**	3.998 *** (0.379)	5.222 *** (0.386)	4.352 *** (0.397)	2.899 ***(0.377)
***n* =**	1711	1711	1712	1712
**Chi-Square=**	483.010 ***	599.118 ***	638.979 ***	593.629 ***
**Cox and Snell=**	0.245	0.294	0.310	0.292
**Nagelkerke=**	0.257	0.306	0.333	0.307

* *p* ≤ 0.05; ** *p* ≤ 0.01; *** *p* ≤ 0.001.

## Data Availability

Due to our Institutional Review Board permissions, we are not at liberty to share raw data.

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
