# Peer review of "Environmental Efficacy, Climate Change Beliefs, Ideology, and Public Water Policy Preferences"

_ijerph, 2021, doi:10.3390/ijerph18137000_

Round 1

Reviewer 1 Report

Dear authors:
Your article has been very interesting to me and I have no contribution to make. Rather, congratulate you on the contribution made.

Best regards

Author Response

Thank you very much for taking the time to review the paper.

Reviewer 2 Report

The research design and conduct are sound and the paper is well-organised, well-written and certainly relevant both for the scientific literature and for policy makers.

Just a couple of in-text questions:

Line 54: ... (often at a rate faster that recharge) ... should be "than"?

Line 155: ... Several studies have explored the connection between personal efficacy ... should be "whether" or "if"?

Author Response

Thank you for taking the time to review the paper. We have made the changes in line 54 to "than" and in line 155 to "whether". Thank you again.

Reviewer 3 Report

Although the authors discuss an interesting issue on environmental efficiency, climate change, ideology, and policy choices, the paper has serious methodological flaws which need to be addressed before considering for publication in the journal, My comments are jotted down below. 

  1. Background and Literature review is well organized and insightful
  2. Methodological discussion is well elaborated.
  3. The abstract lacks the summary of the key results from the study
  4. The importance and novelty of the study are weakly discussed in the introduction. It needs to be improved.
  5. Why OLS rather than maximum likelihood estimations (like ordered probit/logit) since data on most of the variables (including dependent variables) are categorical (nominal, ordinal, and binary) in nature? Such a setup violates the linearity assumption of the OLS estimation approach, leading to unconstrained predicted values when taking the derivatives. In this type of situation, OLS asymptotic properties also do not follow the optimal path, which ultimately questions the point estimates of the variables. Therefore, I strongly recommend authors use discrete choice maximum likelihood models to infer the likelihood/probability for advocating different policies.
  6. The discussion also needs to be further improved by discussing institutional arrangements.

Author Response

Thank you for taking the time to review our paper, and for providing feedback on how it can be improved. We have addressed your suggestions in the revised draft, and will address them point-by-point here.

  1. The abstract now has a summary of key findings;
  2. The novelty of the study is now addressed in the introduction (see lines 61-64);
  3. Regarding why we chose OLS: Many social scientists use OLS for multivariate analyses when using ordinal (rank ordered) data as OLS produces robust results and is very accessible to readers, however there has been much debate on this topic.  As a result, some social scientists have moved to Ordinal Regression (also known as ordinal classification) when conducting multivariate analyses with Likert type scales, such as what was used for the water policies included in this study.  Therefore we replaced the OLS models with Ordinal Regression models in the paper, which lead to nearly identical results as the OLS models.   

Finally, regarding your comment about discussing institutional arrangements, we would be happy to do so, but are a bit unclear as to what you are suggesting. Could you please provide a bit more feedback so we can properly respond to your suggestion?

Thank you again for your time and suggestions. We appreciate the opportunity to improve upon the paper.

Reviewer 4 Report

The paper is well structured and relevant in the literature field investigating climate change in relation with resources consumption. Its position can be recognized in the number of studies aiming to understand how people are keen to accept water related policies or not (thus on citizens science applied to climate change issues). It is in fact focused on investigating, through a survey, the positions of citizens in the United States (especially in the western side) in relation with the most common water policies.

The contribution is well designed, structured and the arguments are well presented and clearly expressed. Tables are also useful for a deep comprehension of the authors' position and findings. However, few improvements can make the contribution clearer and more understandable. I suggest to accept the paper with minor revisions as following.

The Introduction is very well structured and clear. However, a small amount of revisions can make it clearer:

  • I would suggest to add a reference in line 56
  • Lines 51 to 56. It can be useful to add some data supporting the extension of the phenomenon described, in particular what is the rate of groundwater pumping which is faster than recharge? If you have those data, the point will be better clarified. Anyway I suggest to add at least a reference on this point.
  • Line 60, I suggest to anticipate the extension of the survey already here, just saying the total number of surveys you sent.
  • On the content point of view, I suggest to very briefly add more arguments on why you choose four as a case study, in relation with other potential countries. This can also be addressed on the Method paragraph.

Background section:

  • Line 65: the note should report also the extended name of the countries and not only the Acronyms
  • Lines 82-88 are very well written but a map or graph or infographic could ease its readability (this is just a suggestion).
  • In the "State Water Profiles" section a brief overview of the policies related with water should be assessed. The paper is in fact lacking a short description of them. You refer to them several times but they are never explicitly reported. I think that this section could be the appropriate one. You describe in fact very well how water is used and the extension of the problem in each state but not the ongoing policies.

The Literature Review is very well conceived, I do not have any suggestions here.

The Methods and Data section is also well addressed.

The Results section is also properly addressed. I have few suggestions:

  • Table 3 should start on the following page to guarantee its readability

Also the Discussion section of well designed and appropriately described.

Author Response

Thank you for taking the time to review the paper and providing suggestions for improvement. We have addressed your suggestions both in the revised draft and below.

  1. Reference for line 56: (line numbering has changed) we have added a reference and a bit more context in line(s) 47-52.
  2. Lines 51-56 more data/addition of a reference: we have added more information and provided a reference (lines 47-52) to address the issue of groundwater pumping at a faster rate than recharge.
  3. Line 60 include the total number of surveys: we put in the total number of surveys on line 59 (n = 4,695)
  4. Provide an argument for the selection of four states: we have now added context to this in the methods section. It reads, "The four Western states included as case studies were selected because of their similar histories, policy-making institutions, ecosystem services, and their historic exposure to water shortages and drought. All four of the states are currently experiencing water shortage issues and drought leading to state-wide debates about appropriate water policies to address the issues".
  5. Line 65, the note should report the extended name of the states: we have avoided the use of acronyms and included full name of the states (lines 72-73).
  6. A map of graph or infographic to add context to lines 82-88: we have now added a figure (Figure 1) illustrating current drought conditions.
  7. "State Water Profiles" section provide a description of water policies: we have included more detail on each state's water management strategies to provide better context of the policies currently guiding water management (lines 138-147).
  8. Table 3 should start on the following page: we have moved tables to fit on one page.

Thank you again for your time and suggestions.